



# Measurement report: Vertical profiling of particle size distributions over Lhasa, Tibet: Tethered balloon-based in-situ measurements and source apportionment

Liang Ran[1], Zhaoze Deng[1], Yunfei Wu[1], Jiwei Li[1], Zhixuan Bai[1], Ye Lu[1,2], Deqing Zhuoga[3], Jianchun
Bian[1,2,4]

[1]Key Laboratory of Middle Atmosphere and Global Environment Observation, Institute of Atmospheric Physics, Chinese
Academy of Sciences, Beijing, 100029, China

[2]College of Earth and Planetary Sciences, University of Chinese Academy of Sciences, Beijing, 100049, China

[3]Tibet Institute of Plateau Atmospheric and Environmental Science, Lhasa, 850000, China

[4]College of Atmospheric Sciences, Lanzhou University, Lanzhou, 730000, China

*Correspondence to*: Zhaoze Deng (dengzz@mail.iap.ac.cn) and Liang Ran (shirleyrl@mail.iap.ac.cn)

**Abstract.** In-situ measurements of vertically resolved particle size distributions based on a tethered balloon system were

carried out for the first time in the highland city of Lhasa over the Tibetan Plateau in summer 2020, using portable optical

counters for the size range of 0.124~32 μm. The vertical structure of 112 aerosol profiles was found to be largely shaped by

the evolution of the boundary layer (BL), with a nearly uniform distribution of aerosols within the daytime mixing layer and a

sharp decline with the height in the shallow nocturnal boundary layer. During the campaign, the average mass concentration

of particulate matters smaller than 2.5 μm in aerodynamic diameter ($PM_{2.5}$) within the BL was around 3 μg m[-3], almost four

times of the amount in the free troposphere (FT), which was rarely affected by surface anthropogenic emissions. Though there

was a lower level of particle mass in the residual layer (RL) than in the BL, a similarity in particle mass size distributions

(PMSDs) suggested that particles in the RL might be of the same origin as particles in the BL. This was also in consistence

with the source apportionment analysis based on the PMSDs. Three distinct modes were observed in the PMSDs for the BL

and the RL. One mode was exclusively coarse particles up to roughly 15 μm and peaked around 5 μm. More than 50% of total

particle mass was often contributed by coarse mode particles in this area, which was thought to be associated with local dust

resuspension. The mode peaking over 0.5~0.7 μm was representative of biomass burning on religious holidays and was found

to be most pronounced on holiday mornings. The contribution from the religious burning factor rose from about 25% on non-

holidays to nearly 50% on holiday mornings. The mode dominated by particles smaller than 0.3 μm was thought to be

associated with combustion related emissions and/or secondary aerosol formation. In the FT coarse mode particles only

accounted for less than 10% of the total mass and particles larger than 5 μm were negligible. The predominant submicron

particles in the FT might be related to secondary aerosol formation and the aging of existed particles. To give a full picture of

aerosol physical and chemical properties and better understand the origin and impacts of aerosols in this area, intensive field

campaigns involving measurements of vertically resolved aerosol chemical compositions in different seasons would be much

encouraged in the future.

## 1 Introduction

Lhasa, the provincial capital of Tibet, lies almost in the heart of the sparsely populated Tibetan Plateau, which is the highest

plateau in the world with an average altitude of approximately 4320 m (Zhang et al., 2021). As a result of its unique topography,

the Tibetan Plateau plays a vital role in East Asian Summer Monsoon, thereby in the regional and global climate (He et al.,

2019; Chiang et al., 2020). As the most urbanized and populated highland city in Tibet, Lhasa is quite suitable for exploring

the impact of anthropogenic activities on atmospheric components over this remote region.

In the past decade, the population in Lhasa has increased from about 0.55 million in 2010 to nearly 0.87 million in 2020,

according to the newly published data from the Seventh National Population Census

(http://www.xizang.gov.cn/zwgk/zfsj/ndtjgb/202105/t20210520_202889.html, last access on 18 June 2021). Rapid

urbanization and economic growth, also the ever thriving tourism, have given rise to a marked increase in residential, industrial

and traffic emissions. As a religious and cultural center, the widespread religious burning was also an important source for

various air pollutants (Cui et al., 2018b; Wei et al., 2019a). In the low oxygen-containing atmosphere (molecules per volume)

at an altitude of about 3650 m, incomplete combustion could further enhance emissions of air pollutants such as black carbon

and organic aerosols. The probable photochemically active atmosphere due to strong solar radiation at such a high altitude and

low latitude might facilitate the formation of secondary aerosols and gaseous pollutants like ozone (Lin et al., 2008), though

the extent to which secondary production is enhanced remains uncertain considering pressure-dependent reaction rate constants

(Atkinson, 2000). In addition, the topography of Lhasa being located in the Lhasa River Valley and surrounded by high

mountains prevents effective dilution of air pollutants. Consequently, serious air pollution comparable to the level in more

urbanized and developed cities at lower altitudes has been observed (Ran et al., 2014; Duo et al., 2018; Li et al., 2019). However,

it was demonstrated that air quality in Lhasa has been slightly improved in 2017 compared with previous years, due to the

implementation of Action Plan on Prevention and Control of Air Pollution all over China since 2013 (Yin et al., 2019).

Previous studies on aerosols in Lhasa were quite limited. One particular focus was the chemical composition of single particle

(Zhang et al., 2001a; Duo et al., 2015), bulk aerosol (Cong et al., 2011; Gong et al., 2011; Liu et al., 2013; Chen et al., 2018)

and size-segregated aerosols (Zhang et al., 2001b; Wan et al., 2016; Cui et al., 2018a; Wei et al., 2019a, 2019b) near the ground.

Results from these studies revealed that vehicular exhaust, religious activities involved incense burning and biomass burning,

and the suspension of mineral dust were major sources. It was also pointed out from the results of the backward trajectory

analysis that local emissions dominated during the monsoon season (Wei et al., 2019b). Aerosol optical properties such as

multi-wavelength aerosol absorption near the ground (Zhu et al., 2017) and aerosol optical depth (Bai et al., 2000; Zhu et al.,

2019) have also been investigated. However, the particle size distribution, as a critical microphysical property, has rarely been

studied. Using a portable optical particle counter, Cui et al. (2018b) measured particle size distributions (14 size bins) within

the range of 0.14~3 μm for three weeks in summer 2016 in urban Lhasa and emphasized the significance of religious burning

to fine particles. In-situ measured vertical profiles of particle size distributions with a rather coarse size resolution were

obtained from three balloon measurements in 1999 over Lhasa, but were only used to explore an enhancement in fine particle

concentrations near the tropopause (Tobo et al., 2007). The vertical structure of particle size distributions and its temporal

variation within the planetary boundary layer has not yet been examined. To get a better knowledge on the current status of

aerosol microphysical properties in Lhasa and to advance our understanding on the influence of anthropogenic activities, in-

situ measurements of vertically resolved particle size distributions with a high vertical resolution are much needed.

In this study, the vertical structure and temporal variability of aerosol profiles were explored, based on in-situ measurements

of particle size distributions within 1 km above the ground at a suburban site in Lhasa, using optical counters attached to a

tethered balloon. Source apportionment based on measured particles size distributions was further performed to identify

possible sources for aerosols in different layers.

## 2 Measurements

### 2.1 The field site

The city of Lhasa lies in an east-west oriented valley along the Lhasa River, surrounded by mountains up to an elevation of

about 5500 m (Fig. 1a), as shown by SRTM data V4 from Jarvis et al. (2008). The field site is on the Najin campus of Tibet

University (29.64 °N, 91.18 °E) in a suburban area of Lhasa, approximately 5 km to the east of the urban centre (Fig. 1b). As

the commercial and religious centre, the urban area features crowded streets and heavy traffic, partly due to the thriving tourism

and partly due to frequent religious activities. Religious burnings are constantly practiced and especially vigorous on religious

ceremonies in the Potala Palace and the neighbouring famous temples, such as Jokhang Temple and Ramoche Temple, which

are all located in this area and attended by many worshippers and tourists throughout the year. Comparatively, the suburb is

thinly populated with much less traffic. Thus emissions from residential, vehicular and religious sources are expected to be

much weaker in the surrounding area of the site than in the downtown.





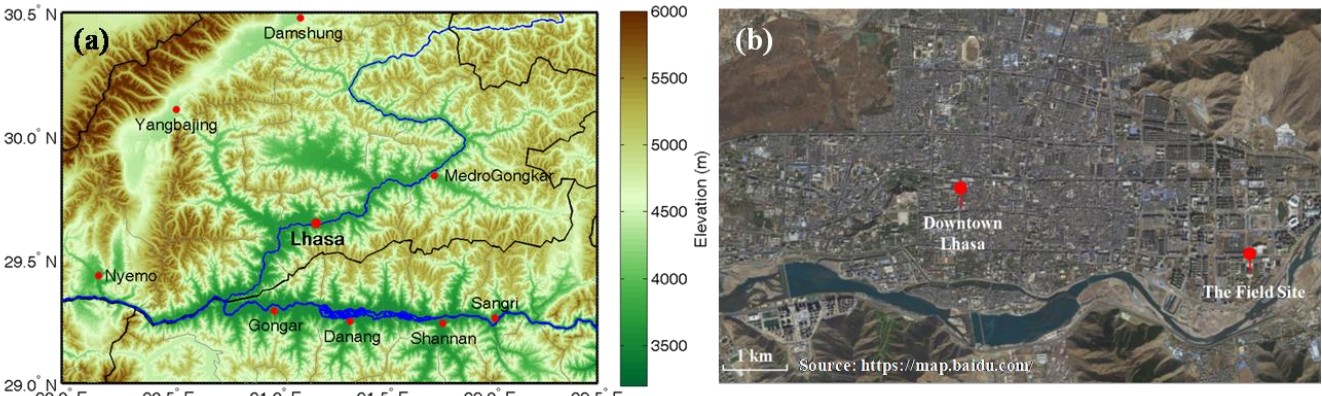

**Figure 1 The topographic map of the Lhasa River Valley (a) and the city map of Lhasa (b).**

## 2.2 Instruments and data

Measurements were conducted in an open space, being kept about 90 m away from the nearest building, in order to ensure the safety of launching the tethered balloon system. The 60 m³ helium-filled tethered balloon, with a payload capacity of roughly 10 kg at an elevation of 3650 m, was operated by an electric winch that controlled the ascent and descent rate of the fish-shaped balloon. It was scheduled to launch the balloon to at least 800 m above the ground every 3 hours from 07:00 to 22:00 (in Beijing time) during 8-28 August 2020. The actual number of launches and the actual maximum height reached by each flight were subject to unavailable flight permits and unfavorable weather conditions (precipitation and/or strong winds). In total, 56 launches were successfully performed for the entire campaign (Fig. S1a). Most flights of the tethered balloon reached above 600 m, with 3 launches reaching 450~500 m, 15 reaching 500~600 m, 17 reaching 600~700 m, 15 reaching 700~800 m, and 6 reaching 800~850 m. In particular, strong winds were often encountered in the late afternoon, resulting in a lack of observations during this time period (Fig. S1b).

A light-weighted optical counter, the Portable Optical Particle Spectrometer (POPS, Handix Scientific), was attached to the tethered balloon for in-situ measurements of vertical profiles of atmospheric aerosols. Details about the principle on which the instrument operates were given in Gao et al. (2016). The POPS was calibrated before the campaign using ammonium sulfate



aerosols (for diameters smaller than 1 μm) and polystyrene latex spheres (for diameters larger than 1 μm). The established

relationship between the scattering signal and the particle size was used to obtain 42 logarithmically equal size bins over the

size range of 0.124~2.55 μm. Ambient aerosols were sufficiently dried before entering the POPS by a homemade silica gel-

filled diffusion drier. The flow rate of the instrument was regularly checked during the campaign and used for data correction.

Particle number size distributions (PNSD, $n(\log D_p)=dN/d\log D_p$) were obtained with a temporal resolution of 1 s, and integrated

over the whole size range to provide total number concentrations ($N_a$). Particle mass size distributions (PMSD,

$m(\log D_p)=dM/d\log D_p$) were calculated from PNSDs assuming spherical particles with a density of 1.7 g cm$^{-3}$ (Sloane et al.,

1991; Tao et al., 2014). Mass concentrations of particulate matters smaller than 1 μm and 2.5 μm in aerodynamic diameter

($PM_1$ and $PM_{2.5}$) were obtained under the assumption that the optically equivalent diameter could be considered equal to

geometric diameter. The effective diameter ($D_e$) was the ratio of the third-order moment to the second-order moment of particle

size distributions. Similarly, the effective diameter for submicron particles ($D_{e,<1μm}$) and for super-micron particles ($D_{e,1~2.5μm}$)

were respectively calculated for particles with the diameter smaller than 1 μm and in the range of 1~2.5 μm.

A portable aerosol spectrometer (Model 11-C, GRIMM Aerosol Technik GmbH & Co. KG) was also employed for 24 launches

to concurrently measure the PNSD for dry particles within the size range of 0.25~32 μm. The measuring time interval was set

to be 6 s, for which 31 size channels were classified. Sizing of aerosols by the instrument was verified before the campaign for

several selected diameters using polystyrene latex spheres. The PNSD measured by GRIMM 11-C was combined with the

PNSD simultaneously measured by the POPS, using weighting factors in the overlapped size range, namely, from the lower

size limit of GRIMM 11-C ($D_{p,L}$) to the upper size limit of the POPS ($D_{p,U}$). The weighting factors for GRIMM 11-C were

derived as $w_{11-C}(\log D_p)=(\log D_p-\log D_{p,L})/(\log D_{p,U}-\log D_{p,L})$, while for the POPS they were defined as $w_{POPS}(\log D_p)=(\log D_{p,U}-$

$\log D_p)/(\log D_{p,U}-\log D_{p,L})$.

Vertical profiles of meteorological parameters, including pressure (MS5540B, Intersema Sensoric SA), temperature ($T$) and

relative humidity (RH) (SHT7x, Sensirion), as well as winds (homemade sensor based on a slotted optical switch (OPB610,

TT Electronics) and an electronic compass module (ZCC211N, Shanghai Zhichuan Electronic Tech Co., Ltd)) were measured

for each flight with a temporal resolution of 1 s. Potential temperature ($\theta$) and specific humidity ($q$), two conservative quantities,

were calculated following the equations in Ran et al. (2016). Besides, surface meteorological parameters in August were

available from an Automatic Weather Station roughly 4.5 km to the northwest of the site in urban Lhasa and were averaged

into hourly data, except that winds were recorded as 2-min averages at the beginning of each hour.

Profiles of aerosol and meteorological parameters were all processed into 10 m-averaged data for the subsequent analysis. A

mixed layer (ML) for daytime profiles or a nocturnal boundary layer (NBL) for nighttime profiles, mentioned altogether as the

boundary layer (BL) in what follows, could be identified for 72 profiles in the dataset of totally 112 profiles. For the remaining

40 profiles, the flight path was entirely within the daytime ML. The height of the BL, denoted as $H_m$ hereafter, was estimated

from profiles of aerosols and meteorological parameters based on the gradient method (Seibert et al., 2000; Kim et al., 2007;

Ferrero et al., 2010; Ran et al., 2016). For profiles with the top of the ML above the flight limit, $H_m$ was taken as the maximum

height reached by the tethered balloon for further statistical analysis. The normalized height ($H_{Nor}$) was then calculated as

135    height/$H_m$-1 (Ferrero et al., 2014). For several profiles, the free troposphere (FT) that was identified as the layer above the BL

without the presence of elevated aerosol layers or the residue from aerosols in the daytime after the collapse of the ML, as well

as the residual layer (RL) could also be identified. In addition, the time mentioned anywhere in this study was Beijing time,

which was around two hours earlier than local solar time.

**2.3 Positive Matrix Factorization (PMF)**

140    A bilinear factor-based receptor model, Positive Matrix Factorization (PMF), was used for source apportionment of particle

mass in this study based upon observed PMSDs (Paatero and Tapper, 1994; Paatero, 1997). By minimizing an objective

function $Q$, a key parameter to review the distribution of the components and to estimate the stability of the solution (Brown et al., 2015), the measured matrix was decomposed into factor profiles ($F$) and factor contributions ($G$) with non-negativity constraints. The input matrix, comprising PMSDs (22 size bins within 0.124~15 µm) averaged for the ML, the RL and the FT,

145 and the lower 10-m averaged PMSDs in the NBL, was used in 20 random PMF runs (using EPA PMF5.0 software) for 2 to 5 factors. In total, there were 48 PMSDs for the ML, 23 PMSDs for the RL and 10 PMSDs for the FT. The components in the PMF runs were separated to keep one signal rather significant from those dominated by noise on account of a criterion of the signal-to-noise ratio (S/N) (Amato et al., 2009). The components with the S/N less than 2 were removed from further analysis, while components with the S/N above 2 were set as "strong" species (Paatero and Hopke, 2003). The stability of converged $Q$

150 values was found for all runs. By estimating diagnostic errors, the 3-factor result was found to be the optimal solution, with all factors mapped in 100% of the Bootstrap run, no swap of Displacement and no swapping case of Bootstrap-Displacement. The results of the scaled residuals for the 3-factor outcome were within an acceptable range of (–3, 3) (Juntto and Paatero, 1994). The measured and PMF-simulated total particle mass concentrations were significantly correlated ($R^2$=0.90).

## 3 Results and discussion

155 ### 3.1 An overview of meteorological conditions

Under the impact of the Asian Summer Monsoon, Lhasa normally experienced a rainy season from May to September (Ding, 2007; Ran et al., 2014). However, in August 2020 when the campaign took place, two periods with distinctly different conditions of water vapor were found, as could be clearly seen from $q$ near the ground measured in urban Lhasa (Fig. 2a). The sharp reduction in $q$ during the daytime of 17 August was an indicator of changing from a humid Period I (averagely 11.1 g

160 kg$^{-1}$) to a relatively dry Period II (averagely 8.1 g kg$^{-1}$). Period I was also characterized as rainy with the precipitation amounting to 78.5 mm and quite cloudy for most of the time, while there was plentiful sunshine and almost no precipitation during Period



II (Fig. 2a). Rainfall mostly occurred in the evening and at night (20:00-08:00), accounting for 84% of the total amount in that

month. The two periods differed not much in $T$, which ranged from 8.8~26.8 °C in Period I and from 8.6~25.0 °C in Period II

with an average of about 17.0 °C for both periods. In contrast, RH averaged nearly 62% in Period I, but only 45% in Period II

(Fig. 2b), corresponding well to the difference in $q$. The temperature shared a similar pattern on each day, except that the

maximum value in the daytime apparently lowered on days with an overcast sky. Unlike the diurnal variation of $T$ being a

peak in the late afternoon (around 18:00) and a valley in the early morning (around 09:00) without much differences between

the two periods, average RH reached a maximum in the early morning of about 79% ±11% for period I and 64% ±6% for Period

II, as well as a minimum in the late afternoon of about 43% ±11% for period I and 28% ±11% for Period II. Surface winds were

largely under 4m s$^{-1}$ and dominated by easterly and westerly winds (Fig. 2c).

Vertical profiles of meteorological parameters observed at the site were categorized into five time periods of the day as give

in Table S1, and were presented along $H_{Nor}$, if either a ML or a NBL could be identified, for a straightforward comparison

between the two periods (Fig. S2). The amount of moisture was obviously higher in Period I than in Period II for all time

periods, being consistent with what had been revealed from continuous surface measurements. The conservative quantities, $\theta$

and $q$, were generally uniform in the ML for daytime profiles, whereas $T$ decreased and RH increased with increasing height,

respectively. For profiles collected during 07:00-08:00 and 20:00-23:00, air temperature inversions in the NBL were

recognized for several cases. Averagely, there was a slight increase in $\theta$ and decrease in $q$ along with increasing height.

Accordingly, wind speed apparently increased with increasing height. A further examination on the probability distributions

of wind speed and wind direction indicated that winds were mainly below 4 m s$^{-1}$ both within and above the BL throughout

the campaign, also being dominated by easterly and westerly winds as near the ground (Fig. S3). Stronger easterly winds

exceeding 4 m s$^{-1}$ occurred above the BL, even up to nearly 7 m s$^{-1}$ in Period II. It should be kept in mind that data points

available for calculating time-period averages at some heights above the BL, corresponding to $H_{Nor}$ larger than 0, might make





up only a part of the total number and might thus introduce misleading details. Caution should thereby be taken about

statistically drawing conclusions from the characteristics of height-normalized profiles across the BL and at different heights

above the top of the BL. The last but not the least, the analysis regarding meteorological parameters, especially winds, was

somewhat limited to measurements under a relatively mild condition when the tethered balloon were able to be launched.

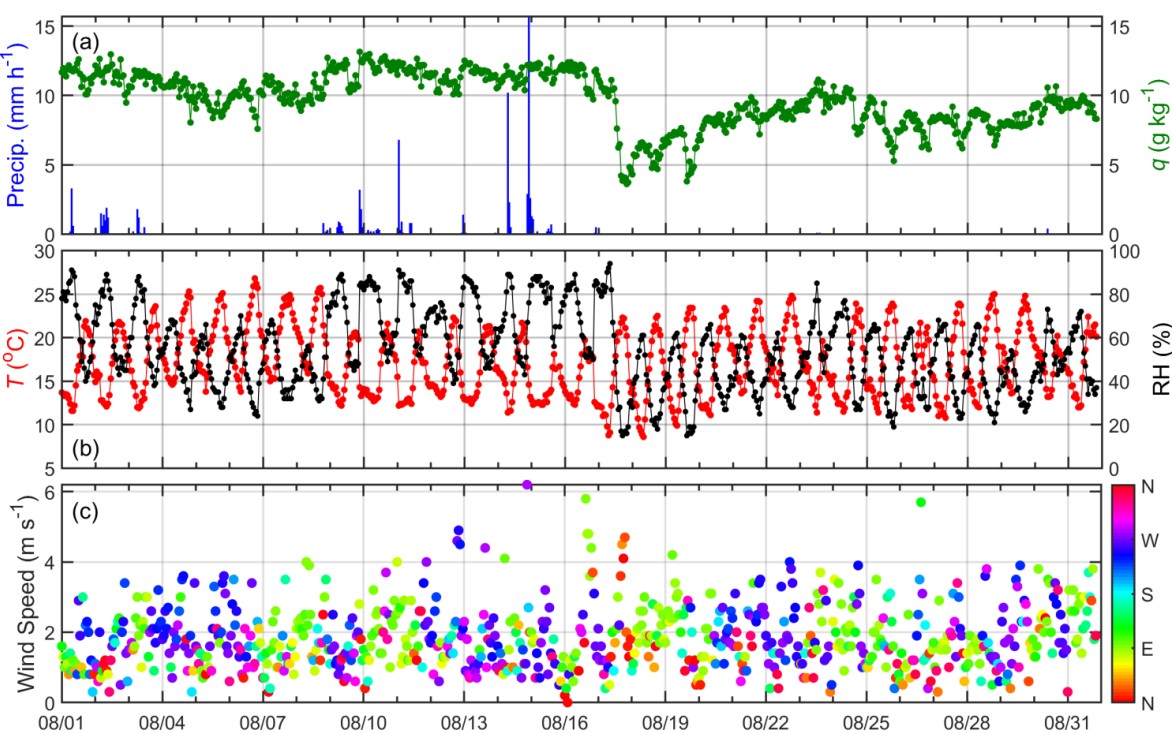

**Figure 2 The time series of (a) precipitation in blue and specific humidity ($q$) in green, (b) temperature ($T$) in red and relative humidity (RH) in black, (c) wind speed and wind direction (in colours), based on hourly data in August from an Automatic Weather**

**Station in urban Lhasa.**





## 3.2 The vertical structure and temporal variability of aerosol profiles

### 3.2.1 Vertical distributions and diurnal variations of aerosol properties

The evolution of the BL was an important influencing factor in shaping the vertical structure of atmospheric aerosols, as being

already pointed out by some previous studies (Ferrero et al., 2010; Rant et al., 2016; Zhang et al., 2017). During the campaign,

heights of the shallow NBL were largely below 200 m around sunrise/sunset and at night (Fig. S4). In Period I, $H_m$ could be

determined for most profiles before midday around 14:00, possibly resulting from the development of the ML being suppressed

under usually cloudy conditions. In the afternoon, the ML often developed higher than the maximum height reached by the

tethered balloon. Unfortunately, intensive observations covering a day from the early morning until the night were unavailable,

either because of unobtainable aviation permission or bad weather. Nevertheless, relatively frequent measurements on 12

August (marked by black cross in Fig. S4) revealed, in a more realistic way as compared with what the overall picture could

tell, the gradual increase of $H_m$ in the overcast morning, from below 100 m before sunrise to about 700 m towards noon. The

depth of the ML on that day was expected to be far above the flight limit of 500 m in the afternoon, since it turned to be quite

sunny and cloudless overhead after midday. As for Period II, the ML should have developed very quickly after sunrise on

those sunny days, thus $H_m$ were often already high above the flight limit when aviation permissions were granted in the late

morning and the afternoon.

Normalized by $H_m$, average vertical profiles of aerosol properties in five time periods of the day were illustrated in Fig. 3,

exhibiting a similarity between Period I and II. Profiles of $N_a$ and particle mass concentrations around sunrise (07:00-08:00)

and since evening (20:00-23:00) resembled an exponential decay in the stable NBL, which favored the accumulation of air

pollutants near the ground. In the daytime, profiles were characterized by a nearly uniform distribution of aerosols within the

ML, except for profiles collected during 08:00-12:00 in Period II with the characteristics of nighttime profiles. This misleading

feature was attributed largely to profiles on religious holidays that were frequently encountered in Period II with the impact of

strong emissions from religious burning in the morning, which will be discussed in detail in Sect. 3.3. With the evolution of the ML, stronger dilution effects led to a decline in surface $N_a$ and particle mass concentrations in the daytime, compared to that in the early morning and at night. However, surface $N_a$ and particle mass concentrations were found to be the highest

during 08:00-12:00 in both periods. The reasons for this phenomenon might be enhanced emissions from more anthropogenic activities after sunrise but still weak dispersion inside a ML, where vertical convection had not been fully developed. Above the BL, the number of data points used for averaging varied with $H_{Nor}$, depending on $H_m$ and the maximum height of each flight. To avoid being misled by insufficient data, averaged data were only adopted at heights where the data availability exceeded 75%. An evident reduction in $N_a$ and particle mass concentrations above the BL was observed by comparison with

that within the BL. For both periods, ratios of $PM_1$ to $PM_{2.5}$ mass concentrations ($PM_1/PM_{2.5}$) were relatively uniform within the BL even for NBL-type profiles, despite distinctly different aerosol amount along the normalized height in the NBL. The daytime $PM_1/PM_{2.5}$ ratios were relatively larger than ratios in the early morning and at night. Accordingly, $D_e$ was found to be relatively smaller during the day.

Table 1 listed particle parameters in the BL, the RL and the FT. For the BL category, averages within the ML and 10-m

averages near the ground for the NBL were adopted. Inside the well mixed layer, the surface level and the ML-averaged level were close to each other. Averaged $N_a$ within the BL was about $338\pm162$ cm$^{-3}$ in Period I and $409\pm201$ cm$^{-3}$ in Period II. The mean $PM_{2.5}$ mass concentration within the BL was $2.7\pm1.7$ µg m$^{-3}$ in Period I, with a fraction of $PM_1$ in $PM_{2.5}$ to be about $81\pm6\%$. In Period II, $PM_{2.5}$ mass concentrations within the BL averaged $3.6\pm2.4$ µg m$^{-3}$, with $PM_1$ accounting for approximately $73\pm9\%$. The lower level of aerosols in Period I might be ascribed to frequent rainfalls and associated efficient removal

processes. Submicron particles took up a smaller proportion in $PM_{2.5}$ during Period II, together with a larger average $D_e$ of 0.35 and a larger average $D_{e,1\sim2.5\mu m}$ of 1.76, indicating more large particles in Period II than in Period I. Though average $D_{e,<1\mu m}$

in the two periods was comparable with each other, a wider range was found in Period II, showing the large variability in

particle size distributions among profiles collected at different time.

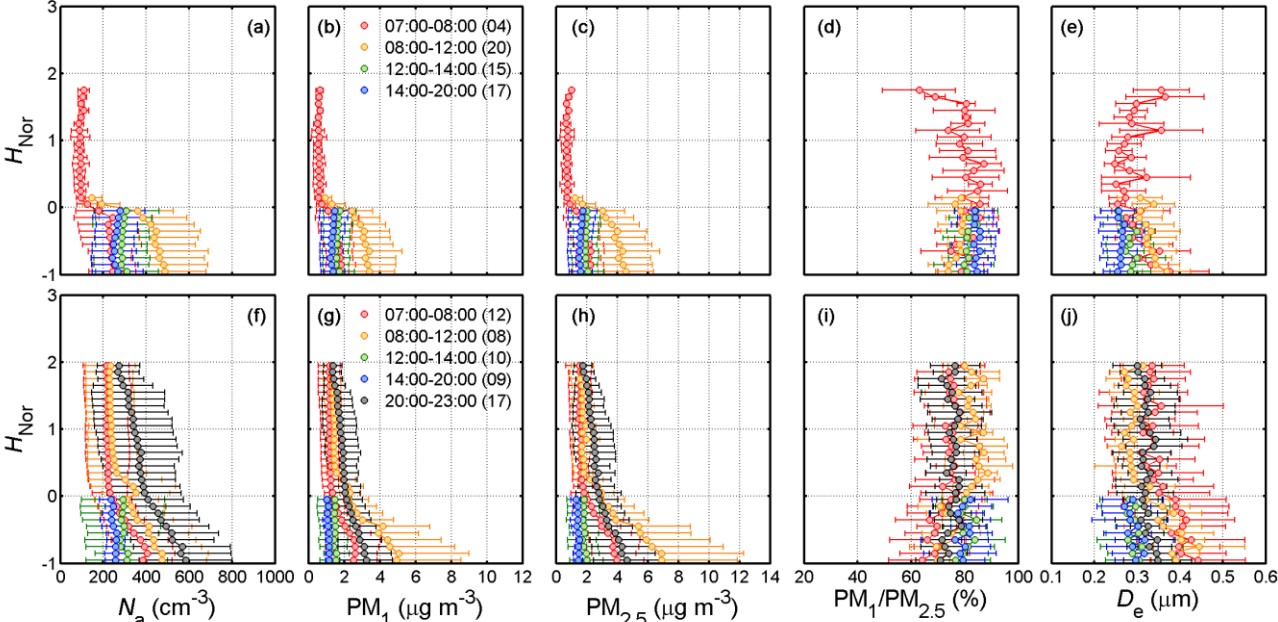

**Figure 3 Average vertical profiles of (a) particle number concentration ($N_a$), (b) PM$_1$ mass concentration, (c) PM$_{2.5}$ mass concentration, (d) the ratio of PM$_1$ to PM$_{2.5}$ mass concentration, and (e) the effective diameter ($D_e$) over the whole size range of the POPS, along the normalized height ($H_{Nor}$) during Period I. Profiles for different time periods are shown in different colours, with dots representing average values and error bars representing standard deviations at each of the $H_{Nor}$. The number of data points used for averaging at each $H_{Nor}$ was more than about 75% of the number of profiles during each time period, as given in parentheses. Average vertical profiles measured during Period II are displayed in the same way in (f)-(j).**

It was found that average PM$_{2.5}$ mass concentrations within the BL during this campaign (3.1 ±2.1 μg m$^{-3}$) were far below what

had been observed near the ground in August 2016 in urban Lhasa, where the high levels often exceeded 20 μg m$^{-3}$ and were

sometimes up to more than 40 μg m$^{-3}$, with the overall average to be 11 ±2.2 μg m$^{-3}$ over the size range of 0.14~3 μm (Cui et

al., 2018b). The significant difference of particle mass concentrations between the previous and the current study might reflect

different emission strengths surrounding the two sites. Being located in the downtown area and adjacent to several temples,

the observational site in the previous study was greatly affected by various strong local emissions, such as traffic, residential



and religious activities-related emissions, while the suburban site in this study was under the influence of much weaker nearby emissions and more likely to represent an average condition over this area. Furthermore, the continuous monitoring of $PM_{2.5}$ mass concentrations from 2013 to 2017 at 6 sites across the city of Lhasa revealed a slight decrease in 2017 (Yin et al., 2019).

This was demonstrated to be partly attributed to measures taken to prevent and control air pollution in Lhasa since 2013. Actually, the location of the site in this study was on the same campus of Tibet University as one site (LHASA-XZ) in Yin et al. (2019). Considering that monthly mean $PM_{2.5}$ mass concentration of all 6 sites for August was reported as about 15 μg m$^{-3}$ and the LHASA-XZ site was not the cleanest one among the others, it might suggest an apparent reduction in $PM_{2.5}$ mass concentrations with the continuing implementation of air quality policies.

Table 1 Particle parameters within the BL, in the RL and in the FT for Period I and II, given as average value ± standard deviation and the range in parentheses. The number of profiles for each category is also given in parentheses in the second row.

| | BL | | RL | | FT | |
|---|---|---|---|---|---|---|
| | Period I (56) | Period II (56) | Period I (0) | Period II (29) | Period I (14) | Period II (10) |
| $N_a$ (cm$^{-3}$) | 338±162 | 409±201 | / | 299±101 | 89±37 | 81±23 |
| | (77~812) | (95~1051) | | (123~575) | (47~156) | (44~112) |
| $PM_1$ (μg m$^{-3}$) | 2.1±1.3 | 2.6±1.7 | / | 1.6±0.6 | 0.5±0.2 | 0.5±0.3 |
| | (0.4~5.5) | (0.5~7.8) | | (0.6~3.1) | (0.3~1.1) | (0.2~0.9) |
| $PM_{2.5}$ (μg m$^{-3}$) | 2.7±1.7 | 3.6±2.4 | / | 2.0±0.7 | 0.7±0.3 | 0.7±0.4 |
| | (0.5~7.8) | (0.6~10.9) | | (0.9~4.0) | (0.3~1.3) | (0.3~1.1) |
| $PM_1/PM_{2.5}$ (%) | 81±6 | 73±9 | / | 78±5 | 79±5 | 76±9 |
| | (62~90) | (48~93) | | (69~86) | (71~87) | (63~89) |
| $D_e$ (μm) | 0.30±0.06 | 0.35±0.09 | / | 0.31±0.03 | 0.29±0.04 | 0.34±0.07 |
| | (0.20~0.47) | (0.21~0.61) | | (0.26~0.37) | (0.24~0.36) | (0.24~0.46) |
| $D_{e,<1\mu m}$ (μm) | 0.23±0.03 | 0.23±0.04 | / | 0.22±0.01 | 0.22±0.01 | 0.23±0.03 |
| | (0.19~0.31) | (0.18~0.35) | | (0.20~0.24) | (0.20~0.24) | (0.18~0.27) |
| $D_{e,1~2.5\mu m}$ (μm) | 1.71±0.15 | 1.76±0.13 | / | 1.72±0.08 | 1.58±0.11 | 1.62±0.18 |
| | (1.34~2.46) | (1.26~2.00) | | (1.56~1.87) | (1.40~1.76) | (1.32~1.78) |

Profiles in Period II were mainly collected in the early morning and at night, when elevated aerosol layers were often encountered, though it was hard to tell whether those were only residual layers that could easily form above the NBL after sunset and remain until the BL evolved to an enough height in the next day, or there could also be contributions from

transported plumes. Here, we generally denoted such elevated layers as residual layers (RL). No RL were observed during

Period I. In Period II, the average particle number and mass concentrations in the RL were respectively around 75% and 55%

of that within the BL. In the FT, average particle number and mass concentrations in the two periods were generally comparable

with each other, and were almost less than 25% of the amount within the BL. In general, fractions of $PM_1$ in $PM_{2.5}$ within the

BL were close to those in the RL and the FT for each period.

Undoubtedly, air quality over the Tibetan Plateau has been increasingly influenced by anthropogenic activities and emissions

during the processes of urbanization and economic growth in the past several decades, especially in relatively thickly populated

cities like Lhasa. However, the level of $N_a$ over Lhasa was still much lower than what has been observed in more populated

and developed regions. Aircraft measurements of particles with the diameter ranging from 0.1 μm to 3.0 μm using a passive

cavity aerosol spectrometer probe (PCASP) over the Loess Plateau, which is adjacent to the Tibetan Plateau, revealed that $N_a$

in the BL often exceeded 1000 cm$^{-3}$ and $N_a$ above the BL reached well above 200 cm$^{-3}$ for most flights during the summer

campaign 2020 (Cai et al., 2021). Influenced more by anthropogenic emissions, $N_a$ over the Loess Plateau were more than 3

times of those obtained within and above the BL over Lhasa. Though a slightly larger average $D_e$ of around 0.4 μm was

observed over the Loess Plateau, no apparent dependence of $D_e$ on altitude was found in both locations. Aircraft measurements

that also employed PCASP in non-winter seasons during 2005 and 2006 over Beijing in the polluted North China Plain showed

that the surface level of $N_a$ ranged from 1000 cm$^{-3}$ (10[th] percentile) to 10000 cm$^{-3}$ (90[th] percentile) and averaged as high as

6600 cm$^{-3}$, more than 10 times of that over Lhasa (Liu et al., 2009). Aerosol profiles measured by a tethered balloon-born OPC

over Milan in Italy, which is also located in a river valley (the Po Valley) but in a region much more industrialized and

populated than the Lhasa River Valley, exhibited an average $N_a$ of 35±2 cm$^{-3}$ in the diameter range of 0.3~20 μm above the

BL in summer (Ferrero et al., 2010). Given the same size range, a lower level of particle number concentrations was found in

the FT over Lhasa, with the average $N_a$ to be 12±8 cm$^{-3}$.

### 3.2.2 Vertically resolved particle mass size distributions

Particle mass size distributions (PMSD) measured by the POPS were categorized into two periods and averaged for three

layers (Fig. 4). In general, size-resolved particle mass concentrations were the highest at all diameters within the BL, while

much lower in the FT. Nevertheless, a similarity was found among all PMSDs. Plainly, a distinct mode below around 0.23 µm

existed for all layers during both periods, though its relative strength to the rest part of the PMSD considerably differed among

different layers and periods. For this mode, more particles were observed towards the lower end of the size range in Period II,

possibly implying stronger secondary formation of fine particles favored by the fine weather during that period. Particles larger

than 1 µm apparently increased with the diameter for all PMSDs, suggesting another mode in coarse particles. The normalized

averages of PMSDs that were merged from the POPS and GRIMM 11-C measurements over the size range of 0.124~32 µm,

which were only available for 48 profiles in Period II, did exhibit a mode peaking around 2.5 µm in the FT and around 5 µm

in the other two layers (Fig. S5). Particles as large as roughly 15 µm were observed within the BL and in the RL. In contrast,

particles with the diameter larger than 5 µm accounted only for a negligible part of the PMSD in the FT. It was also noted,

from both the average PMSD (Fig. 4) and the normalized average PMSD (Fig. S5), that there was a third mode ranging over

0.5~0.7 µm within the BL and in the RL, though relatively less pronounced for the later one, whereas no such an apparent

mode was observed in the FT. These features revealed that under most circumstances particles in the RL were of the same

origin as particles in the BL, whereas particles in the FT should be rarely affected by local anthropogenic emissions near the

ground.

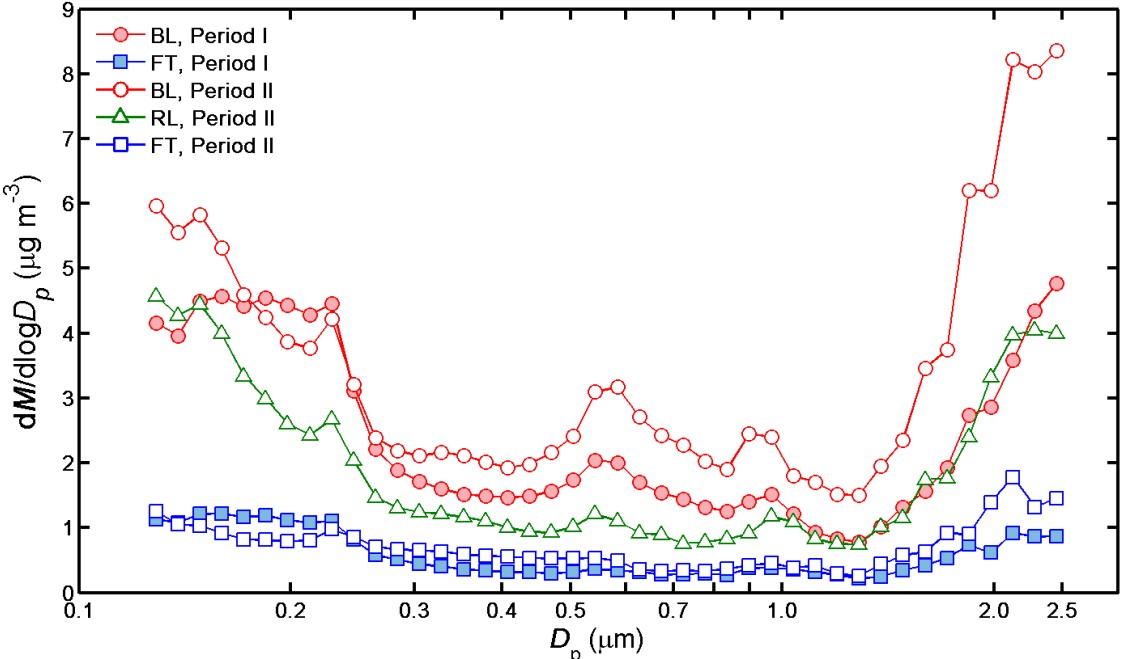

**Figure 4 Average particle mass size distributions (d$M$/dlog$D_p$) for different layers during the two periods as indicated by different markers.**

The vertical distribution of PMSDs within the size range of 0.124~32 μm for each profile in Period II was explored in more detail. It was noteworthy that the mode over 0.5~0.7 μm was found to be more distinct on the mornings of religious holidays compared with that at other times. In order to conveniently make comparisons among different cases, the dataset was grouped into subsets of data collected on non-holidays and holidays. In total, there were 22 aerosol profiles collected on non-holidays and 26 on the four religious holidays (19, 20, 26, 28 August). The subset of data from holidays was further divided into five cases, four of which respectively comprised measurements on the morning of each religious holiday. One case, consisting of data collected on holidays except in the morning, was denoted as holidays* to distinguish from the category named holidays that covered all profiles measured on religious holidays. The average profile of $D_{e,<1\mu m}$ on non-holidays and holidays* were found to be close to each other, with an average of roughly 0.21±0.01 μm and 0.23±0.02 μm through the vertical direction, respectively (Fig. 5a). However, the average vertical distribution of $D_{e,<1\mu m}$ on the morning of each religious holiday, as





separately displayed in different colors, was significantly different from the categories of non-holidays and holidays*. In the

BL, a remarkably larger $D_{e,<1\mu m}$ was observed in the morning on all religious holidays than that on both non-holidays and

holidays*. The largest $D_{e,<1\mu m}$ was nearly 0.35 μm on 19 August, the Shakyamuni Buddha Day and also the beginning of Sho

Dun Festival, one of the most ceremonious traditional festival in Tibet. This could plausibly explain the aforementioned larger

range of $D_{e,<1\mu m}$ averaged within the BL in period II than in Period I, since emissions from religious activities on religious

holidays enlarged the effective diameter and only one religious holiday (8 August) was encountered in Period I. Above the BL,

$D_{e,<1\mu m}$ for the four holiday morning cases generally fell in the range of $D_{e,<1\mu m}$ on non-holidays and holidays, except that

$D_{e,<1\mu m}$ on the morning of 19 August was apparently larger than others. As for coarse particles in the size range of 1~10 μm,

no obvious difference was observed between non-holidays and holidays (Fig. 5b).

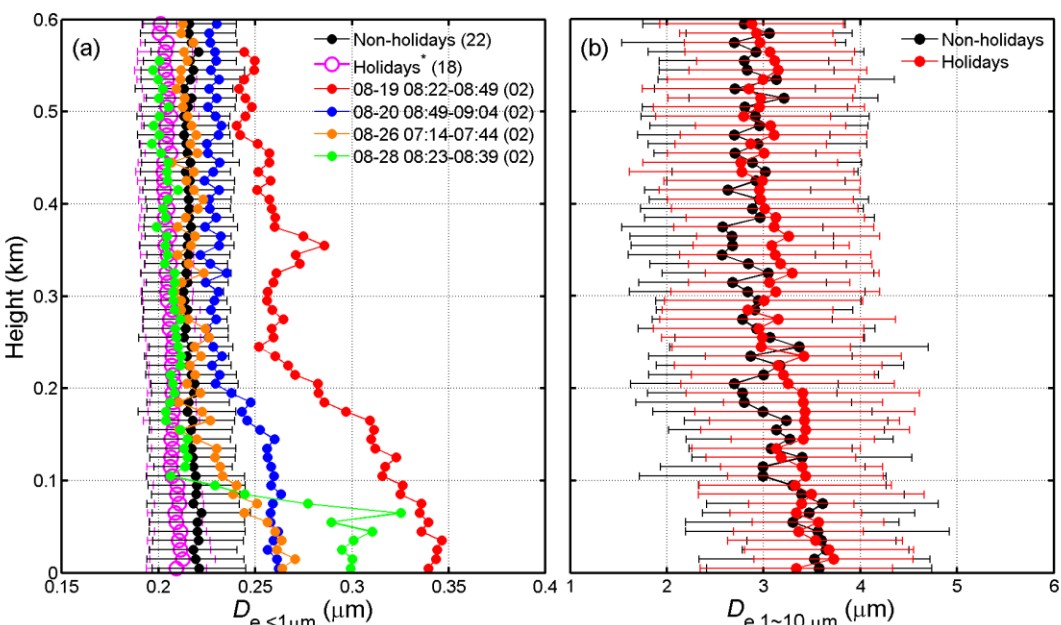


**Figure 5 (a) Average vertical profiles of $D_{e,<1\mu m}$ for different cases: solid black dots for non-holidays, magenta circles for religious holidays but excluding samples in the mornings, various colored solid dots for religious holiday mornings. (b) Average vertical profiles of $D_{e,1\sim10\mu m}$ for non-holidays (black dots) and holidays (red dots). The number of profiles collected during the period of each case was given in the parentheses.**



A further examination was taken on average PMSDs within 50 m above the ground for each category, considering that in the

NBL or in an early morning ML measurements near the ground would better help elucidate the impact religious activities

brought to PMSDs. A marked mode peaking around 0.6 μm and amounting to about 18 μg m$^{-3}$ was observed in the PMSDs on

19 and 28 August (Fig. 6). Though much lower around 5 μg m$^{-3}$ on 20 and 26, the peak of this mode was still almost double

of that on non-holidays and holidays*. The existence of the accumulation mode over 0.5~0.7 μm on religious holidays was

consistent with the findings from surface aerosol measurements in August 2016, and was demonstrated to be characteristic of

emissions from incense burning and biomass burning for religious ceremonies (Cui et al., 2018b and references therein). As a

widespread religious custom, incense burning was commonly performed in the daily life of local people, not only in temples

but also at home. The burning of several types of wood branches and herbs as well as butter lamps in temples was also

traditional religious activities. The PMSDs clearly showed that these emissions were largely enhanced on holidays especially

in the mornings and formed a much strengthened accumulation mode.

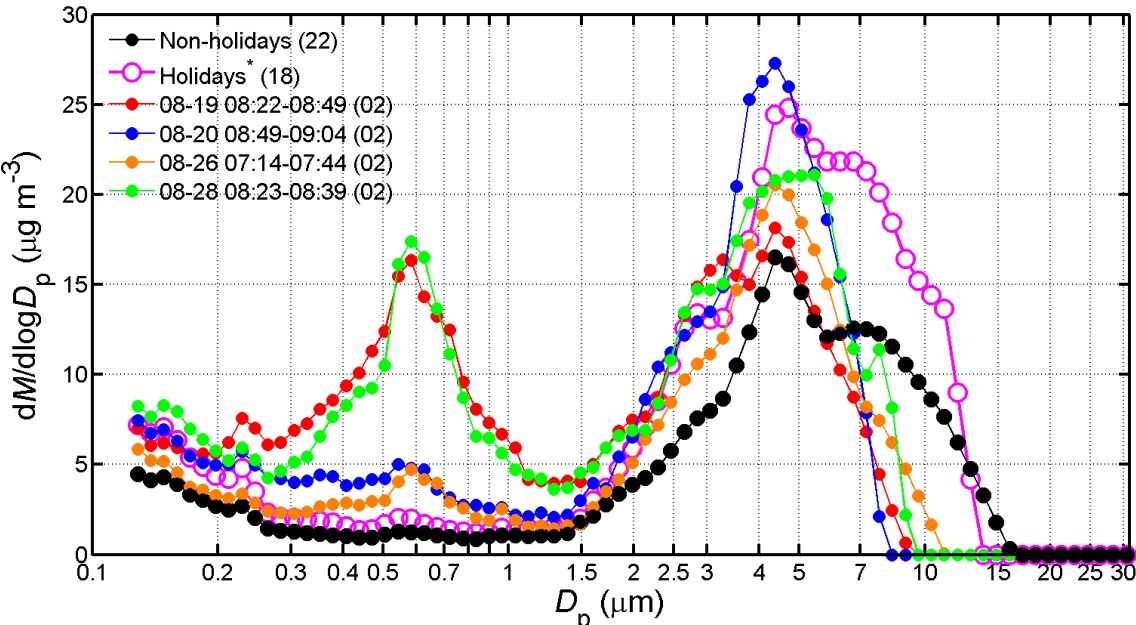

**Figure 6 Average PMSDs within 50 m above the ground for different cases, with the same markers and texts as in Fig. 5.**

### 3.3 Potential sources of particles in different layers

Three source factors were identified from the PMF analysis of the PMSDs combined from the POPS and GRIMM 11-C

measurements in Period II (Fig. 7a). The first factor (factor 1) revealed a potential source that predominantly contributed to

particles smaller than 0.3 μm, probably being associated with local emissions from fossil fuel combustions and/or secondary

aerosol formation. The second factor (factor 2) showed a broad peak over about 0.3~0.7 μm, with also a considerable

contribution from particles with the diameter extending from 0.7 μm to around 2.5 μm. Considering the apparent enhancement

of particle mass in the size range of 0.5~0.7 μm on religious holiday mornings, factor 2 was taken to be mainly representative

of aerosols released from religious activities such as burning incense, cypress branches, herbs, and butter lamp (Cui et al.,

2018b and references therein). The third factor (factor 3) was almost exclusively composed of particles in the coarse mode (>

1 μm). This factor might be attributed to suspended dust particles from unpaved roads, construction sites, and the surrounding

mountains. Indoor particles previously collected at a temple in Lhasa exhibited a bimodal particle mass distribution, with one

peak around 0.4~0.7 μm and one peak in the coarse mode around 5 μm (Cui et al., 2018a). It was speculated that factor 3 might

also involve a certain contribution from religious burning and/or residential biomass burning. However, concentrations

contributed by factor 3 did not increase as expected with the rising concentrations contributed by factor 2 on holiday mornings,

when assuming that coarse mode particles observed at the site were partly generated from religious activities (Fig. S6). Also

residential burning of biomass including cow dung and plants, as an important source in the past for energy, was negligible

nowadays and mostly replaced by a mixture of fossil fuel and renewable energy sources such as solar, wind, geothermal and

hydroelectric power.



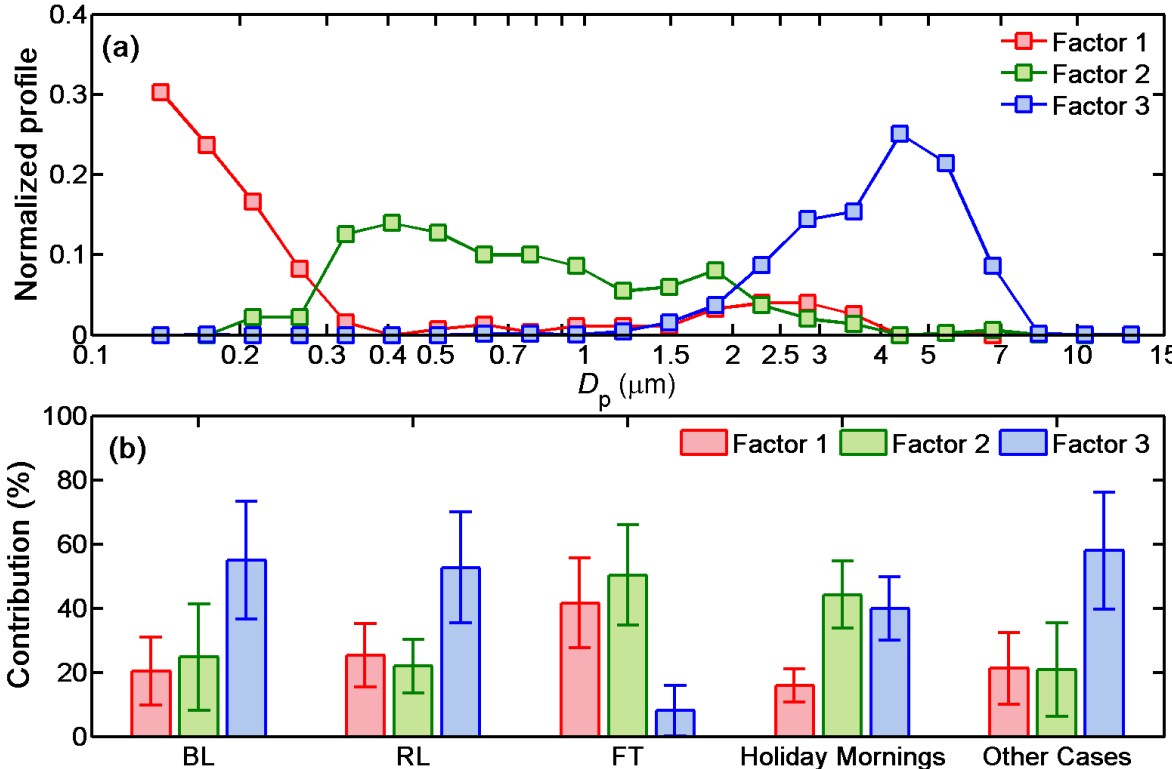

**Figure 7 (a) Normalized factor profiles and (b) averaged contributions of each factor to total particle mass concentrations in the BL, the RL, and the FT in Period II, also for averages in the BL on holiday mornings and other cases, namely, non-holidays and holiday\*, with the error bar indicating the standard deviation.**

On average, particles in the BL and the RL shared a similar makeup of potential sources (Fig. 7b), suggesting that particles in the two layers were probably of the same origin. The contribution by suspended dust particles took up a portion of more than 50%, while the contribution by religious burning was comparable with the contribution by combustion related or secondary formation source with a fraction of roughly 20%~25%. Particularly, the results for the BL were separated into one part on holiday mornings and the other under other conditions, namely, non-holidays and holidays\* as defined above, since particles

in the BL was mostly influenced by emissions near the ground. It was found that on non-holidays and holidays\* the makeup of potential sources was close to that during the whole Period II, while religious burning contributed about 45% on holiday mornings and exceeded the other two factors. However, the relative relationship of the two factors other than religious burning

was similar under both conditions, with the contribution by combustion/secondary formation nearly 40% of the contribution by dust suspension, suggesting these two sources stayed relatively stable during the campaign. An examination on

contributions from the three factors at different time periods on non-holidays revealed the predominant contribution by dust suspension and no obvious diurnal variation of the makeup of potential sources was found. Unlike in the BL and the RL where the significant importance of coarse mode particles was found, the contribution by large particles that were possibly associated with dust suspension was below 10% in the FT. Factor 1 and 2 contributed approximately 40% and 50% in the FT. Although factor 2 was identified as a religious burning factor for the BL and the RL, this might not be the case in the FT, as the

contribution from factor 2 did not differ so much on holiday mornings and at other times. Given the FT was generally much less influenced by emissions near the ground, factor 1 and 2 may be related more to secondary aerosol formation and the aging of existed particles. Measurements of aerosol chemical compositions along the vertical direction in the future might help entangle this issue.

## 4 Summary

In this study, vertical profiles of particle size distributions within 1 km were measured by a POPS and a GRIMM 11-C attached to a tethered balloon in summer 2020 at a suburban site in Lhasa. The variability in the vertical structure and temporal features of parameters such as aerosol number and mass concentrations, the effective diameter and particle mass size distributions were examined. Possible sources for aerosols in different layers were investigated by the PMF analysis.

The vertical distribution of aerosol properties was found to be largely governed by the diurnal variation of the boundary layer.

Generally speaking, aerosols uniformly distributed in the daytime BL and sharply declined in the NBL. Usually average particle number and mass concentrations in the FT were less than 25% of the amount in the BL. For the humid Period I under the influence the Asian Summer Monsoon, a lower level of aerosols and also less large particles in the BL were found than

that in the relatively dry Period II, possibly due to frequent rainfalls and associated efficient removal by wind. More emissions

from religious activities in Period II might also contribute to the differences between the two periods. In contrast, both the total

amount and the size distribution of particle mass in the FT for the two periods were comparable. Residual layers with elevated

aerosol were often encountered in Period II, with particle mass concentrations averagely around 55% of that in the BL. The

PMSDs in the RL shared a similar pattern as that in the BL, suggesting they were probably of the same origin. The source

apportionment analysis revealed that the factor associated with suspended dust in the coarse mode contributed more than 50%

for the BL and the RL. However, a distinct peak over 0.5~0.7 μm was observed on religious holiday mornings, the same time

periods when the contribution by the identified religious burning factor with a broad peak over about 0.3~0.7 μm was found

to exceed the contribution by dust suspension. In the FT, submicron particles dominated and the contribution by coarse mode

particles was below 10%. More investigations based on ground-level long-term monitoring and vertical measurements of

aerosol chemical and physical properties will help better identify the origin and evolution processes of aerosols in this area.

**Data availability**

All data presented in this paper can be accessed by contacting corresponding authors, Zhaoze Deng (dengzz@mail.iap.ac.cn)

and Liang Ran (shirleyrl@mail.iap.ac.cn).

**Author contribution**

JB and LR proposed the study. LR, ZD and ZB designed and conducted the field campaign. YL participated in the field

campaign. LR and ZD processed the data. JL and YW performed the Positive Matrix Factorization modelling. DZ provided

surface meteorological data. LR visualized the data and wrote the manuscript. ZD and YW participated in several discussions

and provided valuable suggestions. All authors reviewed the manuscript carefully.

**Acknowledgements**

This research was funded by the second Tibetan Plateau Scientific Expedition and Research Program (2019QZKK0604). This research was also supported by National Natural Science Foundation of China (NSFC) under grant no. 91837311 and
42061134012. We are grateful to Yong Wang, Hanze Yu and Qi Li for their assistance in launching the tethered balloon. We also thank Tibet University for providing the location for the campaign and all the support.

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
