# Peer review of "Measurement report: Vertical profiling of particle size distributions over Lhasa, Tibet: Tethered balloon-based in-situ measurements and source apportionment"

_Atmospheric Chemistry and Physics, 2021_

## Author Comment (AC1)

Response to referee #2:

**General comments**

**This measurement report is clearly presented and includes valuable observations and methods, important for monitoring and understanding aerosol sources and variability in urban and suburban air quality. Through regular tethered balloon measurements, a substantial number of vertical profiles of aerosol size distribution have been collected in the boundary layer, occasionally reaching into the free troposphere. Three distinct particle size modes are attributed to different sources: local fossil fuel combustion emissions and secondary aerosol formation (< 0.3 μm), aerosol produced in religious activities involving burning incense and other materials (0.3 – 2.5 μm), and coarse mode dust (>1 μm). Relative and absolute weighting of these size modes in the boundary layer were attributed to both meteorological (precipitation, RH, etc.) and anthropogenic causes (in particular, religious burning practices). Free tropospheric conditions appear to be decoupled from the boundary layer, with an occasional residual layer between the BL and FT.**

**The work presented here is appropriately submitted as a Measurement Report, though direct links for access to data (per ACP policy) appear to be missing at this time.**

We highly appreciate the referee's valuable comments and instructive suggestions. We have addressed each comment as below and corresponding revisions have been made in the manuscript. We have also added a direct link to the data presented in this study and revised the section of "Data availability" as "The data in this study can be publicly accessed via https://doi.org/10.5281/zenodo.6374312 (Ran et al., 2022)."

**Specific comments**

**Please address the following technical comments to help clarify and improve the manuscript:**

■ **The two POPS calibration materials (ammonium sulfate and PSL) have different refractive indices. What assumptions are being made about atmospheric aerosol optical properties for measurement with the POPS? Sizing on any such optical particle counter is sensitive to selection of refractive index. Non-monotonic Mie surfaces may influence reported size distributions in certain size regions. This can show up as relatively sharp features in size distributions like the peak seen at 230 nm or 500-900 nm in Figure 4. See Gao et al., 2016 (Figure 8 in particular) for potential influences of refractive index mismatch between calibration and measured material. High resolution binning can also introduce erroneous peaks**

**in distributions, again particularly in regions of Mie resonances or "flat" portions of the Mie curve. Would your analysis change if either/both the 230 nm and 500-700 nm peaks in Figure 4 were due to instrumental settings and assumptions, rather than real features?**

We appreciate the referee's valuable comment. We agree with the referee that sizing is sensitive to the selection of the refractive index and using two calibration materials with different refractive index for particle sizing could introduce misleading interpretation of the data measured by an optical counter such as POPS. We intended to report ammonium sulfate-based PNSDs, as the refractive index of ambient dry particles is closer to that of ammonium sulfate (Shingler et al., 2016) rather than polystyrene latex sphere (PSL), though PSL is widely used in the calibration of OPCs. However, without available methods to classify super-micron ammonium sulfate particles, a calibration curve combined from experimental responses of ammonium sulfate particles at submicron sizes and PSL at super-micron sizes was adopted in the original manuscript. As found in Gao et al. (2016), there are strong oscillations in theoretical response curves for polystyrene latex sphere (PSL) and dioctyl sebacate (DOS) when particle diameters are larger than 600 nm, leading to unrealistic particle sizing. Similar oscillations were also found for ammonium sulfate particles based on our calculation. Therefore, the response curve for converting the optical signal to the particle size above 600 nm should be smoothed to be monotonically increasing. In the revised manuscript, we re-calibrated POPS data using a combined calibration curve from the experimental response of ammonium sulfate particles for diameters smaller than 600 nm and the smoothed theoretical response of ammonium sulfate particles for diameters larger than 600 nm. By using the new calibration method, PNSDs were obtained assuming only the refractive index of ammonium sulfate. Though new results were quantitatively different, but the changes in numbers and plots were rather small and qualitative conclusions remained the same. The manuscript has been revised accordingly. We have also revised the description of POPS calibration as below.

Page 6, Line 101: The POPS was calibrated by establishing a relationship between the scattering signal and the particle size before the campaign. Both polystyrene latex sphere (PSL) with known sizes and ammonium sulfate particles with sizes selected by a differential mobility analyzer were employed. Though the experimental responses of the two calibration materials generally agreed well with the simulated theoretical responses, both theoretical response curves were found highly oscillatory above the particle size of 600 nm (Gao et al., 2016). Considering that the refractive index of ammonium sulfate is closer to ambient dry aerosols than PSL (Shingler et al., 2016), a combined calibration curve from the experimental response of ammonium sulfate particles for diameters smaller than 600 nm and the smoothed theoretical response of ammonium sulfate particles for diameters larger than 600 nm was used to obtain 42 logarithmically equal size bins over the size range of 0.124~2.55 μm.

We agree with the referee that high solution binning might result in fake peaks and valleys in PNSDs due to the non-monotonic response. We performed a comparison among average PMSDs for different layers using POPS data generated with a resolution of 8, 16 and 32 bins per magnitude. The plots below showed a peak in the size range of 0.5~0.7 μm for all resolutions. However, the resolution of 8 bins per magnitude was too coarse to sufficiently display the characteristics of the PMSDs. As for the small peak around 230 nm indicated by the referee, it was indeed more pronounced in PMSDs based on a resolution of 32 bins. Since no solid evidence or physical/chemical reasons to prove such a peak was real, we speculated this might be resulted from uncertainties in the measurement or calibration. We added in the manuscript a discussion as below.

Page 18, Line 315: Plainly, a distinct mode below 0.3 μm existed for all layers during both periods... It was noteworthy that the small peak around 0.23 μm might be resulted from uncertainties in the measurement or calibration. Thus, caution should be taken when drawing specific conclusions.

[Figure]

Plots of average particle mass size distributions for different layers during the two periods using POPS data with a resolution of (left) 8 bins, (middle) 16 bins, (right) 32 bins per magnitude.

References:

Gao, R. S., Telg, H., McLaughlin, R. J., Ciciora, S. J., Watts, L. A., Richardson, M. S., Schwarz, J. P., Perring, A. E., Thornberry, T. D., Rollins, A. W., Markovic, M. Z., Bates, T. S., Johnson, J. E., and Fahey, D. W.: A light-weight, high-sensitivity particle spectrometer for PM2.5 aerosol measurements, Aerosol Sci. Technol., 50, 88-99, doi: 10.1080/02786826.2015.1131809, 2016.

Shingler, T., Crosbie, E., Ortega, A., Shiraiwa, M., Zuend, A., Beyersdorf, A., Ziemba, L., Anderson, B., Thornhill, L., Perring, A. E., Schwarz, J. P., Campazano-Jost, P., Day, D. A., Jimenez, J. L., Hair, J. W., Mikoviny, T., Wisthaler, A., and Sorooshian, A.: Airborne characterization of subsaturated aerosol hygroscopicity and dry refractive index from the surface to 6.5 km during the SEAC4RS campaign, J. Geophys. Res. Atmos., 121, 4188-4210, doi:10.1002/2015JD024498, 2016.

- **A figure illustrating the combined POPS + GRIMM 11-C distribution and the validity of the weighting and combination method of the individuation distributions would be fitting, perhaps in the SI.**

We have added in the supplement a figure to illustrate how the PNSD and the PMSD measured by POPS and GRIMM 11-C were combined using the weighting factors for each instrument. We have revised the manuscript accordingly.

Page 8, Line 2: An example of combining the PNSD from POPS and GRIMM 11-C and the PMSD from POPS and GRIMM 11-C using the weighting factors for each instrument was illustrated in Fig. S2.

[Figure]

Figure S2 An example of combining the PNSD from POPS and GRIMM 11-C (b) and the PMSD from POPS and GRIMM 11-C (c) using the weighting factors ($w_{\log D_p}$) for each instrument (a).

- **Is there a reason residual layers were only seen in Period II, not in Period I? Also, how was the upper limit of the RL defined, to distinguish from the FT?**

The residual layer (RL) usually occurrs after the collapse of the planetary boundary layer (PBL) in the evening and might last until the early morning on the next day. The residues would usually be gradually mixed down into the
5  PBL along with the evolution of the PBL after sunrise in the morning. As described in the manuscript, profiles in Period II were mainly collected in the early morning and at night, when the maximum height reached by the tethered balloon exceeded the top of the PBL and the RL could be observed if it existed. During Period I, all profiles were collected before the evening and a few in the early morning. The FT was identified for 14 profiles but no RL was identified. For four profiles collected in the morning, there was actually a layer above the PBL and beneath the FT
10  with $N_a$ to be a little higher than that in the FT but still very low (~100 cm$^{-3}$). Since it was insufficient for us to confidently decide whether it was the RL or still a part of the FT, we did not include them into either the RL or the FT. Another reason that we did not identify them as the RL was that the statistical results might be misleading if we included these cases, especially when the determination was not solid. We thank the referee for this question to make us realize that it would be better to give a clearer and more precise description of this issue. We have revised
15  the manuscript as below.

Page 16, Line 286: In total, the RL was identified for 29 profiles in Period II, whereas none was identified in Period I. A layer above the PBL and beneath the FT was observed for four profiles collected in the morning in Period I, with $N_a$ to be a little higher than that in the FT but still very low (~100 cm$^{-3}$). Since it was insufficient to decide whether the layer was the RL or still a part of the FT, also it would be misleading if they were used to represent the
20  average condition of the RL during Period I, they were identified as neither the RL nor the FT.

When the maximum height reached by the tethered balloon exceeded the top of the identified RL, the upper limit of the RL was determined by subjectively analyzing vertical distributions of $N_a$ and PMSDs. A sharp reduction in $N_a$ could be found from the RL to the FT, where the average $N_a$ was at least less than half of that in the RL. Both $N_a$ and PMSDs greatly varied across the top of the RL and gradually became almost unchanged after entering into
25  the FT.

**Technical corrections**

**I believe the following grammatical changes will clarify the authors' intended meanings.**

- **Line 20: "in consistence" change to "consistent"**

We have revised the manuscript accordingly.

- **Line 57: "religious activities involved incense burning…" change to "religious activities which involve incense burning"**

We have revised the manuscript accordingly.

10 - **Lines 159 & 160: "averagely" change to "average"**

We have revised the manuscript accordingly.

- **Line 177: "Averagely" change to "On average"**

We have revised the manuscript accordingly.

15

- **Line 185: "The last but not the least" change to "Finally" or "Last but not least"**

We have revised the manuscript accordingly.

- **Lines 193 & 194: "as being already pointed out by" change to "as has been pointed out by"**

20 We have revised the manuscript accordingly.

- **Line 259: "evolved to an enough height in the next day, or there could also be contributions" change to "had evolved to sufficient height the next day, or if there could also be contributions"**

We have revised the manuscript accordingly.

- **Line 263: "were almost less than" change to "were generally less than"**

We have revised the manuscript accordingly.

5 - **Line 366: "was" change to "were"**

We have revised the manuscript as "It was found that on non-holidays and holidays* contributions of potential sources were…"

- **Line 378: "entangle" change to "disentangle"**

10  We have revised the manuscript accordingly.

---

## Author Comment (AC2)

Response to referee #1:

**The manuscript is well organized and should be accepted after a minor revision, especially related to the method section.**

We greatly appreciate the referee's valuable comments and helpful suggestions. We have addressed all comments as below and revised the manuscript accordingly.

**Here below the specific comments:**

■ **Line 15, page 1: "boundary layer (BL)", better "planetary boundary layer (PBL)"**

We have changed "boundary layer (BL)" to "planetary boundary layer (PBL)" through the entire manuscript as the referee suggested.

■ **Line 17, page 1: "aerodynamic diameter (PM2.5) within the BL was around 3 µg m$^{-3}$"; please add the confidence interval at 95%.**

To give a better description, we have revised the manuscript as below.

Page 1, Line 16: During the campaign, mass concentrations of particulate matters smaller than 2.5 µm in aerodynamic diameter ($PM_{2.5}$) within the PBL ranged from 0.5 to 12.0 µg m$^{-3}$, with an average and standard deviation of 3.4±2.3 µg m$^{-3}$ …

■ **Line 22 page 1: "One mode was exclusively coarse particles up to roughly 15 µm and peaked around 5 µm", please detail the cound mean diameter and the geometric mean standard deviation of each mode.**

The description of this coarse mode as well as the other two modes in the abstract intended to summarize the characteristics of average PMSDs for the PBL, the RL and the FT, followed by estimating contributions of possible sources to each mode from PMF analysis, not the exact fitted modes in the PMSD for each layer.

■ **Line 80, page 4: It should be nice to add (in supplementary) a picture of these religious burning events.**

Pictures of religious burnings in temples as mentioned in section 2.1 on page 4 were not available, since it is not allowed to take photos in temples.

- **Line 100-103: Using different calibration standards (ammonium sulfate and polystyrene latex sphere) would affect the detected size distribution. Indeed the two standards are characterized by different refractive indexes thus generating different responses of the optical particle counter to the ambient aerosol. Please add a discussion on this methodology point considering the expected size distribution distortion.**

We agree with the referee that the two calibration standards differ in the refractive index, which could lead to different optical responses. We intended to report ammonium sulfate-based PNSDs, as the refractive index of ambient dry particles is closer to that of ammonium sulfate (Shingler et al., 2016) rather than polystyrene latex sphere (PSL), though PSL is widely used in the calibration of OPCs. However, without available methods to classify super-micron ammonium sulfate particles, a calibration curve combined from experimental responses of ammonium sulfate particles at submicron sizes and PSL at super-micron sizes was adopted in the original manuscript. Referee #2 also raised concerns using two calibration materials. We noticed in Gao et al. (2016) that theoretical response curves for PSL and dioctyl sebacate (DOS) were highly oscillatory above the particle size of 600 nm, leading to unrealistic particle sizing. Similar oscillations were also found for ammonium sulfate particles based on our calculation. Therefore, the response curve for converting the optical signal to the particle size above 600 nm should be smoothed to be monotonically increasing. In the revised manuscript, we re-calibrated POPS data using a combined calibration curve from the experimental response of ammonium sulfate particles for diameters smaller than 600 nm and the smoothed theoretical response of ammonium sulfate particles for diameters larger than 600 nm. By using the new calibration curve, PNSDs were obtained assuming only the refractive index of ammonium sulfate. Qualitative conclusions based on results using the new dataset remained the same, though quantitatively there were small changes in numbers and plots. We have revised the manuscript accordingly. We have also revised the description regarding POPS calibration as below.

Page 6, Line 101: The POPS was calibrated by establishing a relationship between the scattering signal and the particle size before the campaign. Both polystyrene latex sphere (PSL) with known sizes and ammonium sulfate particles with sizes selected by a differential mobility analyzer were employed. Though the experimental responses of the two calibration materials generally agreed well with the simulated theoretical responses, both theoretical response curves were found highly oscillatory above the particle size of 600 nm (Gao et al., 2016). Considering

that the refractive index of ammonium sulfate is closer to ambient dry aerosols than PSL (Shingler et al., 2016), a combined calibration curve from the experimental response of ammonium sulfate particles for diameters smaller than 600 nm and the smoothed theoretical response of ammonium sulfate particles for diameters larger than 600 nm was used to obtain 42 logarithmically equal size bins over the size range of 0.124~2.55 μm.

References:

Gao, R. S., Telg, H., McLaughlin, R. J., Ciciora, S. J., Watts, L. A., Richardson, M. S., Schwarz, J. P., Perring, A. E., Thornberry, T. D., Rollins, A. W., Markovic, M. Z., Bates, T. S., Johnson, J. E., and Fahey, D. W.: A light-weight, high-sensitivity particle spectrometer for PM2.5 aerosol measurements, Aerosol Sci. Technol., 50, 88-99, doi: 10.1080/02786826.2015.1131809, 2016.

Shingler, T., Crosbie, E., Ortega, A., Shiraiwa, M., Zuend, A., Beyersdorf, A., Ziemba, L., Anderson, B., Thornhill, L., Perring, A. E., Schwarz, J. P., Campazano-Jost, P., Day, D. A., Jimenez, J. L., Hair, J. W., Mikoviny, T., Wisthaler, A., and Sorooshian, A.: Airborne characterization of subsaturated aerosol hygroscopicity and dry refractive index from the surface to 6.5 km during the SEAC4RS campaign, J. Geophys. Res. Atmos., 121, 4188-4210, doi:10.1002/2015JD024498, 2016.

- **Line 104: Please add the relative humidity ensured by the diffusion dryer.**

The sample air was dried by the diffusion dryer filled with silica gel. Previous tests showed that roughly 75% and 50% of the water vapor could be removed by the diffusion dryer in summer and winter, respectively. As a matter of fact, the relative humidity will be further lowered inside the instrument, due to the temperature difference between the ambient condition and instrument chambers. The combined effects could ensure the relative humidity inside the POPS to be well below 20% and 30% in summer and winter, respectively. We have added a description of the diffusion dryer in the manuscript as below.

Page 6, Line 113: Previous tests of the diffusion dryer showed that roughly 75% and 50% of the water vapor could be removed after the drying in summer and winter, respectively. As a matter of fact, the relative humidity will be further lowered inside the instrument, due to the temperature difference between the ambient condition and instrument chambers. The combined effects resulted in the relative humidity inside the POPS to be well below 20% and 30% in summer and winter, respectively.

- **Line 106-107: A direct conversion of PNSDs into PMSDs can bring to some errors as the density of the fine and coarse mode can be different. Please compute and add to the paper also the Volume Size Distribution analysis (PVSD) for scientific consistency. Consider that the POPS suffer of a truncation error and an ambient aerosol density application will bring to a mass concentration underestimation.**

We agree with the referee that the density of fine particles might differ from that of coarse particles. We also agree with the referee that converting PNSDs into PMSDs based on the assumption of a constant density over a range of particle sizes might bring some uncertainties in the calculated PMSDs. However, by assuming a fixed density of 1.7 g cm$^{-3}$, the analysis of PMSDs in this study could actually be taken to some extent as the analysis of particle volume size distributions (PVSDs), as the two distributions shared similar shapes and variations with a difference only determined by a fixed factor.

The upper limit size of the POPS in this study was 2.55 μm, equivalent to the aerodynamic diameter of 3.32 μm assuming a density of 1.7 g cm$^{-3}$. Therefore, the underestimation of PM$_{2.5}$ mass concentrations due to size truncation of the instrument was negligible. We agree with the referee that the actual density of ambient aerosols, which was not available in this study, could be different from the assumed value. However, results regarding PMSDs/PVSDs should not qualitatively be much affected by the assumption of a fixed density. We have thereby added a discussion on possible uncertainties in the manuscript as below.

Page 7, Line 121: Though the assumption of a fixed density for particles with different diameters as well as the difference between the actual density of ambient particles and the assumed one might introduce some uncertainties in the calculated PMSDs, results regarding PMSDs would qualitatively not be much affected.

- **Line 110-112, page 6: "Mass concentrations of particulate matters smaller than 1 μm and 2.5 μm in aerodynamic diameter (PM1 and PM2.5) were obtained under the assumption that the optically equivalent diameter could be considered equal to geometric diameter". Here there is a big mistake. Please consider that the PM1 and PM2.5 are defined in function of an aerodynamic diameter and of an efficiency collection curve in inertial impactors (or similar cut-off devices such as cyclones). The authors should add at least a comparison between their PM1 and PM2.5 estimation and the same data obtained using gravimetric samplers.**

We appreciate the referee's valuable comment. We considered the optically equivalent diameter equal to the geometric diameter and converted the geometric diameter to the aerodynamic diameter assuming the density to be

1.7 g cm$^{-3}$. However, no aerosol penetration curves were considered for the calculation of PM$_1$ and PM$_{2.5}$ mass concentrations in the original manuscript. We agree with the referee that the calculation of PM$_1$ and PM$_{2.5}$ mass concentrations should take into account aerosol penetration curves of impactors as they are defined. We recalculate PM$_1$ and PM$_{2.5}$ mass concentrations, adopting the penetration curve of SCC-2.229 cyclone at 16.7 lpm for

5 calculating PM$_1$ mass concentrations (https://bgi.mesalabs.com/wp-content/uploads/sites/35/2015/02/scc_btr-2.229.pdf, accessed on 15/03/2022) and VSCC-2.946 cyclone at 16.7 lpm for calculating PM$_{2.5}$ mass concentrations (https://bgi.mesalabs.com/wp-content/uploads/sites/35/2015/02/vsccref6-2.946.pdf, accessed on 15/03/2022). According to comments of both referees, a new calibration curve was used to obtain particle sizes and corresponding number concentrations. The final results regarding PM$_1$ and PM$_{2.5}$ mass concentrations in the revised

10 manuscript were based upon the newly derived PNSDs and the new calculation method involving penetration curves. Though we did not have simultaneous measurements from gravimetric samplers for comparisons, we compared PM$_1$ and PM$_{2.5}$ mass concentrations calculated from the previous method without penetration curves and the new method with penetration curves. Averagely, mass concentrations using the new method changed 1.1±0.8% for PM$_1$ and -0.2±1.6% for PM$_{2.5}$ relative to that using the previous method. Conclusions drawn from comparisons

15 between PM$_1$ and PM$_{2.5}$ mass concentrations in this study and in other studies were not affected, when using the new method instead of the previous one. We have added more details on obtaining the aerodynamic diameters and a description of aerosol penetration curves in the manuscript as below.

Page 7, Line 125: …and the aerodynamic diameter could be converted from the geometric diameter with a density of 1.7 g cm$^{-3}$. The penetration curve of SCC-2.229 cyclone at 16.7 lpm (https://bgi.mesalabs.com/wp-

20 content/uploads/sites/35/2015/02/scc_btr-2.229.pdf, accessed on 15/03/2022) was adopted for calculating PM$_1$ mass concentrations and the penetration curve of VSCC-2.946 cyclone at 16.7 lpm (https://bgi.mesalabs.com/wp-content/uploads/sites/35/2015/02/vsccref6-2.946.pdf, accessed on 15/03/2022) was adopted for calculating PM$_{2.5}$ mass concentrations.

25 ■ **Line 113-114: Was the GRIMM 11-C equipped with a dryer. If yes was it deployed on the balloon?**

The GRIMM 11-C was also equipped with the homemade silica gel-filled diffusion drier as was POPS and attached to the tethered balloon for 24 launches. To better clarify the condition, we have revised the manuscript as below.

Page 7, Line 134: A portable aerosol spectrometer (Model 11-C, GRIMM Aerosol Technik GmbH & Co. KG), equipped with the homemade silica gel-filled diffusion drier, was also attached to the tethered balloon for 24 launches to concurrently measure the PNSD for dry particles within the size range of 0.25~32 μm.

■ **Line 133-134: Are there any radiosoundings in Lahsa that should be used to unravel the situations in which the top of ML was above the top of the profiles?**

There were no radio soundings when flights of the tethered balloon were totally within the ML to unravel such situations.

■ **Line 149: "strong species"; maybe better "strong aerosol types"? Species remind to chemical speciation which is not present in this work.**

We have revised the manuscript accordingly.

■ **Line 163: better "ranged from" … "to"…**

We have revised the manuscript accordingly.